# Hydrogen-induced tunable remanent polarization in a perovskite nickelate

Yifan Yuan ®[1,10] ✉, Michele Kotiuga ®[2,10] ✉, Tae Joon Park ®[3,10] ✉, Ranjan Kumar Patel[1], Yuanyuan Ni[4], Arnob Saha[5], Hua Zhou ®[6], Jerzy T. Sadowski ®[7], Abdullah Al-Mahboob[7], Haoming Yu[3], Kai Du ®[8], Minning Zhu[1], Sunbin Deng[3], Ravindra S. Bisht[1], Xiao Lyu[9], Chung-Tse Michael Wu ®[1], Peide D. Ye ®[9], Abhronil Sengupta[5], Sang-Wook Cheong ®[8], Xiaoshan Xu ®[4], Karin M. Rabe[8] & Shriram Ramanathan[1] ✉

Materials with field-tunable polarization are of broad interest to condensed matter sciences and solid-state device technologies. Here, using hydrogen (H) donor doping, we modify the room temperature metallic phase of a perovskite nickelate $NdNiO_3$ into an insulating phase with both metastable dipolar polarization and space-charge polarization. We then demonstrate transient negative differential capacitance in thin film capacitors. The space-charge polarization caused by long-range movement and trapping of protons dominates when the electric field exceeds the threshold value. First-principles calculations suggest the polarization originates from the polar structure created by H doping. We find that polarization decays within ~1 second which is an interesting temporal regime for neuromorphic computing hardware design, and we implement the transient characteristics in a neural network to demonstrate unsupervised learning. These discoveries open new avenues for designing ferroelectric materials and electrets using light-ion doping.

Since the discovery of spontaneous polarization in Rochelle salt[1], much effort has been made to design new materials with macroscopic polarization at room temperature. Electrets are materials that have quasi-permanent electric fields at their surfaces or electric polarization. Ferroelectric materials as one class of electrets with permanent electric dipole, have been developed into practical devices[2], such as ferroelectric random-access-memory and actuators[3]. Another class of electrets, space-charge electrets, in which the polarization originates

from an imbalance of charge due to charge trapping or injection, offers promising applications in xerography, powder coating, and electrostatic precipitation[4,5]. To date, much effort has been made to design materials exhibiting and enhancing ferroelectric polarization or space-charge polarization. For example, bulk and 2D ferroelectric materials are discovered by exploring structures that break centrosymmetry in the polarization direction[6]. Furthermore, multi-component superlattices are proposed to enhance ferroelectric polarization through

[1]Department of Electrical & Computer Engineering, Rutgers, The State University of New Jersey, Piscataway, NJ, USA. [2]Theory and Simulation of Materials (THEOS), National Centre for Computational Design and Discovery of Novel Materials (MARVEL), École Polytechnique Fédérale de Lausanne (EPFL), Lausanne, Switzerland. [3]School of Materials Engineering, Purdue University, West Lafayette, IN, USA. [4]Department of Physics and Astronomy, University of Nebraska–Lincoln, Lincoln, NE, USA. [5]School of Electrical Engineering and Computer Science, The Pennsylvania State University, University Park, State College, PA, USA. [6]X-ray Science Division, Advanced Photon Source, Argonne National Laboratory, Lemont, IL, USA. [7]Center for Functional Nanomaterials, Brookhaven National Laboratory, Upton, NY, USA. [8]Department of Physics and Astronomy, Rutgers, The State University of New Jersey, Piscataway, NJ, USA. [9]School of Electrical and Computer Engineering and Birck Nanotechnology Center, Purdue University, West Lafayette, IN, USA. [10]These authors contributed equally: Yifan Yuan, Michele Kotiuga, Tae Joon Park. ✉e-mail: yifan.yuan10@gmail.com; mkotiuga@materialsdesign.com; tjp059@gmail.com; shriram.ramanathan@rutgers.edu

interfacial coupling[7]. On the other hand, space-charge electrets with a net electrostatic charge, e.g., ionic electrets, result from injecting electrons or ions on the surface or at deeper sites[5]. Hence, the quest to discover novel materials with spontaneous polarization is an active area of research and the application space is continuously evolving with societal grand challenges. At the same time, it is important to understand the polarization mechanisms and distinguish them from leakage current artifacts that might interfere with the measurements.

Here, using proton doping realized via hydrogen intercalation in NdNiO$_3$ (NNO), as a simple and scalable approach, we demonstrate H-NdNiO$_3$ (H-NNO) thin films with both intrinsic dipolar polarization and space-charge polarization starting from electron-conducting metallic NNO thin films. The hydrogen acts as an electron donor thereby residing as protons in the interstitial sites and the electron is anchored to the Ni-O hybridized states strongly suppressing the conductivity. While the metal-to-insulator transition in NNO resulting from H doping has been studied[8–11], to our knowledge there have been no reports yet on the polarization properties. Proton doping can render a polar structure that produces a spontaneous dipole moment within the lattice while greatly suppressing electrical conductivity. The long-range movement of protons and trapping further enables a large space-charge polarization. We find that the polarization is metastable and relaxes within ~1 s, which is an interesting temporal regime to design artificial neurons for neuromorphic computing hardware[12]. Moreover, the observed transient differential negative capacitance demonstrates the existence of switchable polarization in H-NNO.

## Results

### Characterization of proton doped NdNiO$_3$ thin films

Epitaxial NNO films were grown by sputtering on (001)$_c$-oriented Nb:SrTiO$_3$ (Nb:STO) substrates (subscript "c" denotes cubic structure). After the deposition of Pd electrodes on the NNO film, the sample was annealed at 200 °C in H$_2$/N$_2$ (5/95) atmosphere for 0.5 hours. The optical images of the pristine NNO and H-NNO-based device (Fig. S1a, b) reveal that, for the H-NNO films, the areas around Pd electrodes are heavily doped. The thickness of H-NNO measured by X-ray reflectivity is ~148 nm (Fig. S1c). To further examine the doping effect on the NNO film, synchrotron X-ray absorption measurements, and spatially resolved synchrotron X-ray microdiffraction were performed while scanning the focused beam across the two Pd electrodes. The map in Fig. 1a was acquired with the X-ray absorption ratio of $I_A/I_B$ at two energies 853:854.2 eV near the Ni $L_3$-edge, which is a suitable proxy to spatially resolve the effect of proton doping on the electronic properties. Figure 1b shows the normalized Ni $L_3$-edge X-ray absorption spectra (XAS) at different micro-regions illustrated in Fig. 1a and from the pristine NNO film (control sample), which are ascribed to a transition from Ni 2p to 3d states at around 853 eV[13]. Compared with the calculated Ni$^{3+}$ spectrum adopted from ref. 14, the A/B peak feature suggests that most nickel ions in the pristine NNO film are in a Ni$^{3+}$ state. As proton doping into NNO films is via catalytic spillover[15], the region adjacent to the Pd electrodes has a higher concentration of protons than the center area between two electrodes due to the diffusive process, which is clearly illustrated in Fig. 1a. From XAS spectra at different regions of the H-NNO film, the spectral weight at peak B gradually decreases as the proton concentration increases. The resemblance of the spectrum at the heavily doped region with that of NiO from ref. 16 indicates H doping decreases Ni valence gradually from Ni$^{3+}$ to Ni$^{2+}$, which is consistent with literature reports[17].

Micro X-ray diffraction (μXRD) measurements were performed to probe the structural evolution of H-doped nickelates (H-NNO) with scans around Nb:STO (002) peak. Micro diffraction $\theta-2\theta$ scans with focused X-rays across two Pd electrodes are displayed in Fig. 1c as a function of positions. Figure 1d shows the representative scans at different positions. The XRD spectrum of the center region shows a (002)$_{pc}$ NNO peak (defined in a pseudocubic cell), near the (002)

substrate peak, indicating that the NNO film is grown along Nb:STO [001] out-of-plane direction. It can be seen that the (002)$_{pc}$ peak is broadened with the injection of protons and begins to overlap with the substrate peak, which has also been noted in literature[18]. The μXRD of H-NNO at different regions indicates that the injection of protons shifts the (002)$_{pc}$ peak (defined in a pseudocubic cell) towards a low diffraction angle, implying the crystal lattice expands along the out-of-plane direction (c-axis). The largest peak shift occurs in the edge area around Pd electrodes, indicating the edge area has the highest proton concentration. This feature is more obvious in the x-ray diffraction microscopy (XDM) image (Fig. 1e), where a large area of the sample is mapped based on the intensity at L = 1.93, the main peak in H-NNO. The shift of the NNO (002)$_{pc}$ peak was also confirmed by laboratory XRD experiments on a series of H-NNO films with different doping levels (Fig. S2).

Synchrotron reciprocal space mapping (RSM) was performed to further probe the structural evolution of nickelates upon hydrogenation. The RSM (Fig. 1f) around (112) reflections of the film and substrate shows the NNO film peak is in the vicinity of the substrate peak, indicating the oriented NNO film is well strained along the in-plane direction. The measured peak positions, in-plane ($a_{pc}$) and out-of-plane ($c_{pc}$) lattice constants of the NNO film are shown in Table S1 and Fig. S3. This confirms that the in-plane strain, defined as $\varepsilon_\parallel = (a_{pc} - a_{pc,bulk\ NNO})/a_{pc,bulk\ NNO}$, is tensile for the NNO film on Nb:STO substrates with $\varepsilon_\parallel$ of +1.54%. The structural behavior of our NNO films on Nb:STO agrees with previous reports[19,20].

The RSM for the H-NNO film in Fig. 1g shows the L value for the (112)$_{pc}$ peak decreases while the K value remains similar upon hydrogenation, resulting in an increase of $c_{pc}$ by 1.2% and almost no change of $a_{pc}$ (Table S1 and Fig. S3), which has a similar trend reported in literature[19,21]. The dominant change of $c_{pc}$ with hydrogenation can be attributed to in-plane substrate constraint and the out-of-plane direction is allowed to expand.

### Hydrogen-induced remanent polarization

We first set the framework for studying polarization by using a symmetric metal/insulator/metal (MIM) structure shown in Fig. 2a. From the simulation of the electric field profile with COSMOL Multiphysics (Fig. 2a), it is seen that most of the polarization is induced by the out-of-plane electric field due to the small resistivity of 0.003 Ω · cm in the conductive substrate Nb:STO. The electric displacement versus electric field, D-E, hysteresis loops of pristine NNO and H-NNO films are presented in Fig. 2b, c. The elliptical loop for the pristine NNO film (control sample) is the feature of a resistor with the measured resistance of 140 Ω, indicating its metallic phase at room temperature. By proton doping, the shape of D-E loops for H-NNO becomes sloped due to the significantly increased resistance in H-NNO as an insulating phase (resistivity ~6 × 10$^8$ Ω·cm). This is consistent with previous research that proton doping can increase the resistance of perovskite rare-earth nickelates up to several orders of magnitude, such as SmNiO$_3$[8,9], LaNiO$_3$[22], and NdNiO$_3$[10,11,15]. When the electric field increases up to a large magnitude, the loop becomes more circular as the device is more lossy at a higher electric field (Fig. S4a). No features of ferroelectric saturation are observed by bipolar D-E loops.

To further understand the polarization behavior, a double-wave method (DWM)[23] or PUND measurement was performed by applying a double-triangular waveform voltage (see Fig. 2d inset for the applied waveform). The first positive half-wave 'P' produces a larger remanent displacement at zero applied fields than the second positive half-wave 'U'. Similarly, the two negative sweeps produce different remanent displacements as well. The difference between the primary and secondary loops produces PUND loops shown in Fig. 2d inset. In comparison, no remanent polarization is observed in the PUND loop for the pristine NNO (Fig. S4b). As the electric field increases, the remanent polarization $P_r$ measured by PUND in H-NNO increases and saturates as

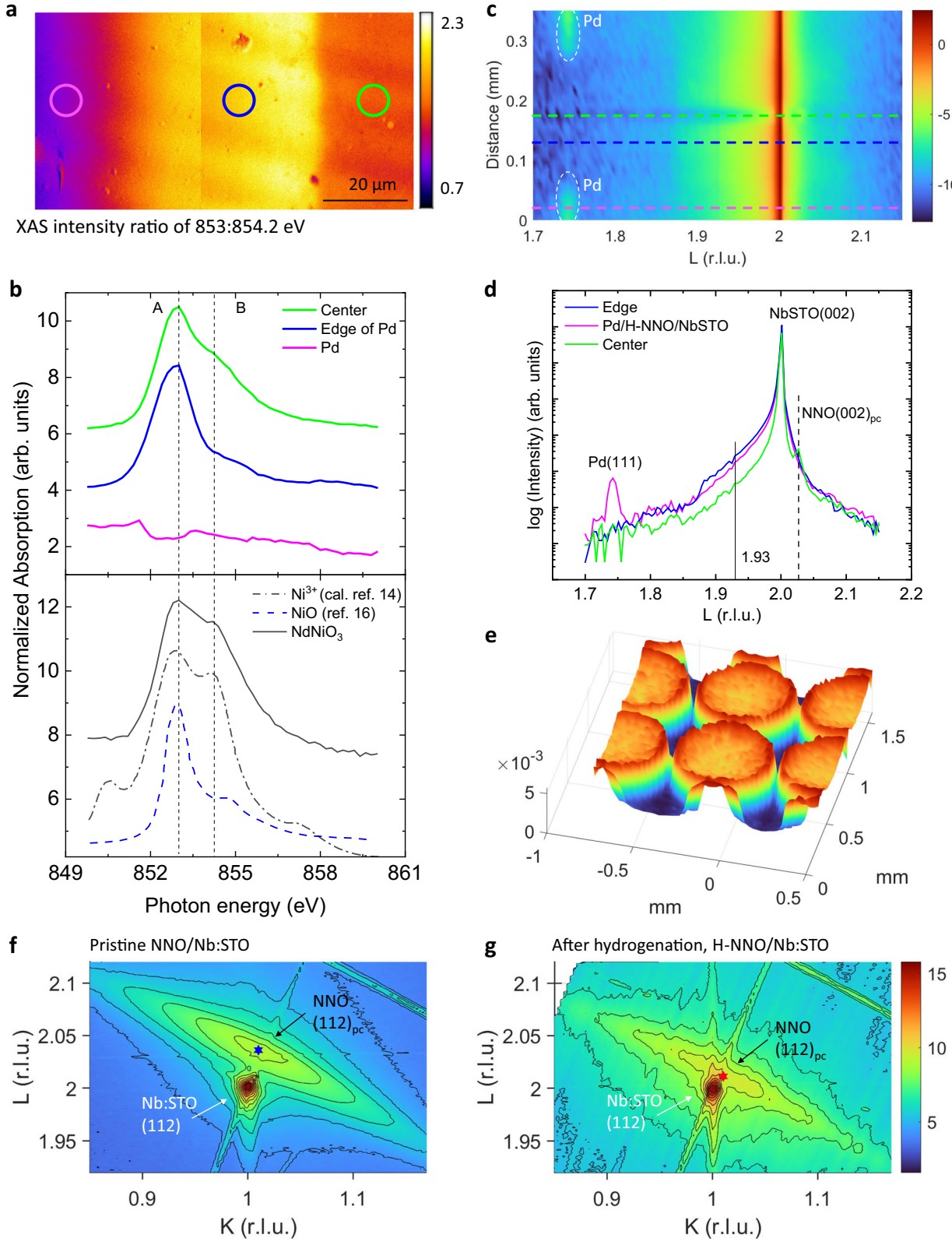

shown in Fig. 2e. These characteristics suggest that the remanent polarization originates from intrinsic dipole switching which is the characteristic of ferroelectrics. Interestingly, when the electric field exceeds ~220 kV/cm (see Fig. S5c), $P_r$ starts quickly rising again with the electric field. This unconventional $P_r(E)$ behavior at high electric fields can be largely attributed to factors other than intrinsic dipole switching, as discussed below.

To demonstrate the behavior of spontaneous polarization $P_s$ with electric field, multiple double-wave measurements were performed with different maximum electric fields and the same field-varying rate ($dE/dt$) (Fig. S5). With the same field-varying rate, the initial part of the primary loops in each measurement is found to coincide well (Fig. S5b, e), excluding experimental artifacts shown. The secondary loops do not overlap in the rising part of each

**Fig. 1 | Structural characterization of pristine NNO and H-NNO films. a** Spatially resolved map of X-ray absorption signal ratio for $E = 853$ eV and 854.2 eV. Each XAS image was obtained over 50-$\mu$m$^2$ area and two images are stitched together. The solid circle indicates the area where local representative XAS spectra in (**b**) were acquired. The color in different areas corresponds to the extent of valence reduction of Ni ions due to the presence of H$^+$. **b** Ni $L_3$ XAS of the H-NNO films on Nb:STO substrates. The Ni-edge XAS spectra of the proton-doped region start to resemble that of NiO. The edge areas around electrodes have low Ni valence due to heavy doping. **c** Synchrotron micro diffraction $\theta-2\theta$ scans around Nb:STO (002) peak with focused X-rays across two Pd electrodes as a function of positions. **d** Several representative XRD spectra were extracted from (**c**) for the H-NNO film.

Nb-doped SrTiO3 substrate Bragg peak is at L = 2. The NNO (002)$_{pc}$ peak is at L = 2.027. **e** The 3D mesh of the XRD signal from L = 1.93 for the H-NNO film over 1.5-mm$^2$ surface indicates the proton doping concentration induced lattice structural contrast and is consistent with the XAS map. The edge areas have the most H-NNO thin film phase due to catalytic hydrogen doping around Pd spots. Reciprocal lattice mapping (RSM) around (112) diffraction peaks of (**f**) pristine NNO and (**g**) hydrogenated H-NNO films on Nb:STO substrates. Panel (**f**) and (**g**) use the same color bar. The hexagrams mark the (112)$_{pc}$ peak positions of the films, which are determined by fitting the contours. The H-NNO films for (**c**) are moderately H-doped at 200 °C and for (**g**) are heavily H-doped at 250 °C (see "Methods"). The color bar in (**c**, **e**, **f**, **g**) is the logarithm of the intensity.

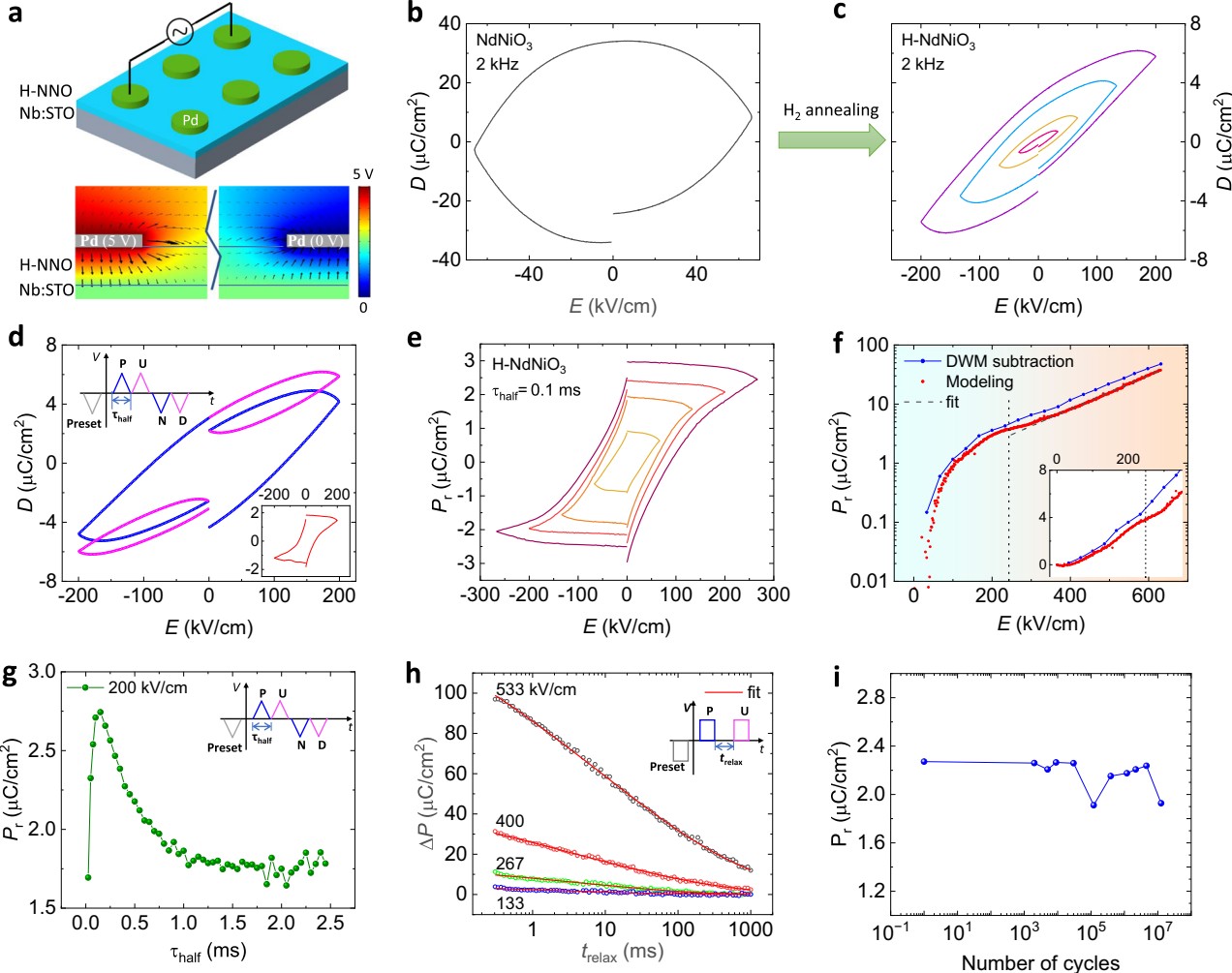

**Fig. 2 | Polarization measurements on H-NNO films. a** The schematic of the H-NNO device and the electric-field distribution simulated with COMSOL. The arrow size denotes the norm of the electric field and the color denotes the potential. **b**, **c** The electric displacement vs. applied electric field measured at 2 kHz for pristine NNO and H-NNO films. **d** The PUND measurement for the H-NNO film. The period of the half wave $\tau_{half}$ is 0.1 ms. Inset is the extracted PUND loop as a function of the electric field from the measurement. **e** PUND loops with different maximum electric fields and the same period of half wave of 0.1 ms. **f** Remanent

polarizaton vs. electric field. $P_r$ in blue dots is obtained from PUND loops and $P_r$ in red dots is from modeling as described in Supplementary Note 1. Inset is an enlarged plot of $P_r$ vs the electrield field ranging from 0 to 350 kV/cm. **g** Remanent polarization as a function of period of half wave $\tau_{half}$ with the same maximum electric field of 200 kV/cm. **h** $\triangle P$ as a function of $t_{relax}$. $\triangle P$ is the difference of the polarization by the pulses P and U. $t_{relax}$ is the time interval between two pulses as shown in the inset. **i** $P_r$ measured by PUND loops versus the number of switching cycles.

measurement, indicating that non-switching loops are affected by the spontaneous polarization, i.e., $P_s$ from switching loops which can suppress non-remanent displacement. For instance, the effect of spontaneous polarization on electric properties has been observed in many aspects such as electric conduction[24], and interfacial band structure[25]. We note that due to the suppressed non-

switching displacement, $P_r$ (or $P_s$) is overestimated by subtracting the secondary loop from the primary loop. To acquire the accurate relationship between $P_r$ and $E$, the device is modeled with three components in a parallel circuit (Fig. S6): a spontaneous-polarization-only capacitor causing hysteresis, a parasitic capacitor $C$, and a nonlinear resistor $R_v$. Both $C$ and $R_v$ are affected by $P_r$,

i.e., total displacement is, $D = P_r(V) + D_C(P_r) + D_R(P_r)$. By deducting the simulated non-remanent displacement, which is detailed in Supplementary Note 1, the extracted $P_r(E)$ is shown in Fig. 2f, red curve. The data in Fig. 2f, plotted in linear scale in the inset, reveal that the modeling method results in the same trend of $P_r(E)$ as the PUND subtraction, but a smaller $P_r$. When the electric field is less than ~243 kV/cm (local minima of $dP_r/dE$), we observe the 'S'-shaped curve of $P_r(E)$ which is a significant characteristic in ferroelectrics, implying the switching of the proton-involved electric dipole within the lattice. At an electric field larger than 243 kV/cm, the induced $P_r$ follows an exponential relationship with the applied electric field. Although many conduction mechanisms exhibit exponential functions with voltage, such as Schottky emission[24,26], electron hopping conduction[27,28], in view of the high mobility of protons[15,29] in NNO thin films, here we ascribe the $P_r(E)$ behavior at high electric fields to charge carrier trapping[30–32] with an ionic hopping mechanism[27,33,34]. In Fig. 2f the black dashed line shows the fit from exponential functions $P_r = P_0(2k_B T)/(k_0 qd) \exp(-\phi_B/(k_B T)) \exp(Eqd/(2k_B T))$, deduced from the function $J = J_0 \exp(-\phi_B/(k_B T) + Eqd/(2k_B T))$, where $P_0, J_0$ is the proportional constant, $J$ is the current density, $\phi_B$ is the potential barrier height, $q$ is the charge on the moving ion, $k_B$ is the Boltzmann constant, $T$ is the temperature, $d$ is the hopping distance. We note that the proton-trapping induced $P_r$ can reach up to 158 μC/cm² (Fig. S8a), which is unusual by compared with other charge-trapping induced polarization, for instance, ~4 μC/cm² for ZrO₂[35]. The estimated $P_r$ due to the intrinsic dipole switching is around 3.7 μC/cm² (Fig. 2f).

## Polarization relaxation dynamics in H-NNO thin films

We carried out further studies of the time/frequency dependence of the $P_r$ values by PUND loops for the same 148-nm-thick H-NNO capacitor (Fig. S8b). The extracted $P_r$ as a function of $\tau_{half}$, the period for one monopolar wave, is shown in Fig. 2g and exhibits a maximum at 0.15 ms. To investigate the precise relaxation process, we applied double pulses, which are schematically shown in Fig. 2h inset, where $\triangle P$ is defined as the difference between the switching and the non-switching displacement. As shown in Fig. 2h and Fig. S9a with normalized $\triangle P$, $\triangle P$ decreases rapidly following an exponential-law dependence on $t_{relax}$ and down to 20% after ~1 s; furthermore, the rate of decay is slower for higher initial values of $\triangle P$ (as shown in Fig. S9a inset,). Similar polarization relaxation or loss phenomena can be seen in ultrathin ferroelectric BaTiO₃ films[36,37] and organic ferroelectric BTA[38] which follows exponential decay. Many ferroelectric materials exhibit polarization relaxations due to the depolarization field. The literature on polarization relaxation is summarized in Supplementary Note 2.

The relaxation behavior can be well described by a stretched exponential function[38], $\triangle P(t_{relax}) = P_0 \exp(-(t_{relax}/\tau)^\beta)$, where $P_0$ is the initial polarization, $\tau$ the characteristic relaxation time and $\beta$ smaller than unity. The fitted parameters are shown in Table S2 where the largest $\tau$ is 8 ms. Note that such a natural polarization relaxation with millisecond timescale is desirable to emulate the leaky-integrate-fire behavior of neurons for neuromorphic computing[39].

The strong polarization relaxation can be related to leakage currents. As shown in Fig. S5, the non-switching $P$ at zero field accounts for 68% of the total $P$, which is much larger than that in normal ferroelectrics, such as 6% in organic ferroelectric MBI[40]. The charge owing to the leakage current will neutralize the polarized charge. On the other hand, protons as dopants in the lattice have high mobility[34], i.e., a small energy barrier, which results in a small threshold of depolarization field due to incomplete charge screening. The endurance measurements up to ten million cycles on the H-NNO capacitor are shown in Fig. 2i. Polarization does not change significantly after 10⁷ cycles which is an encouraging preliminary result.

## First principles calculation of the hydrogen-induced polarization

To understand the origin of polarization switching in H-NNO, we performed density-functional-theory (DFT) calculations, with a Hubbard U correction to investigate the non-polar and polar structures and the resulting spontaneous polarization. The details of calculations are given in Methods. The structure with added hydrogen was optimized while accommodating the experimental epitaxial constraint, i.e., allowing the bulk lattice parameters of NNO to expand in the direction of epitaxial growth. For a non-polar structure, we find that the structural symmetry in NNO is lowered from *Pbnm* (space group #62) to *P2₁/m* (space group #11) by the epitaxial constraint (Fig. 3a, d). By adding four H atoms to various combinations of preferential sites to construct several H-NNO cells, lattice parameters $a = 5.410$ Å, $b = 6.105$ Å, $c = 7.526$ Å with a monoclinic angle $\gamma = 86°$, resulting in a 6.7% increase of the lattice parameter in the epitaxial [001] direction. It is also found that the NiO₆ octahedra are, on average, 12% larger than the pristine NNO. This increase in volume stems from the hydrogen-donated electron localizing on the NiO₆ octahedra resulting in an enclosed Ni²⁺.

Figure 3b, c, and Fig. S11 show the projected density of states (PDOS). We note that as the donor electrons localize on NiO₆ octahedra, there are no hydrogen-like states within 6 eV of the Fermi energy. While the calculated bandgap is always larger than 2.5 eV, we find a variation of ~0.3 eV based on the positions of the hydrogen. The slight differences in the structure and splitting of the peaks in the PDOS, most clearly seen 3-4 eV above the Fermi energy, arise from the local structural differences due to the asymmetric distortion of the NiO₆ octahedra around the hydrogen ions.

To investigate the change in polarization as a function of hydrogen positions, a set of structures was constructed in which the atomic positions smoothly evolve, starting from the non-polar structure and visiting the polar structures. The results are summarized in Table 1. The structures "Polar 1 & 6" shown in Fig. 3e, f are the lowest in energy and are nearly degenerate. The change in polarization can be tracked by the movement of the hydrogen ions. This can result in large changes, on the order of a quantum of polarization, when the hydrogen ions are moved across the cell. This is not surprising as the hydrogen ions are quite mobile in similar materials as they exhibit ionic conductivity at higher temperatures[29].

## Transient negative differential capacitance (NDC) in H-NNO

To validate the hypothesis of spontaneous polarization in H-NNO and explore potential applications, we have measured the charge-switching dynamics in the capacitors. As shown in Fig. 4a inset, square voltage waves are applied to the circuit in which the lossy capacitor is in series with a load resistor of 702 kΩ. The source voltage $V_s$ and the voltage across the capacitor $V_f$ were monitored simultaneously. From Fig. 4b, c, it can be seen that there are two scenarios where the free charge increases (decreases) but the voltage decreases (increases) during the polarization reversal. In Fig. 4c, upon the switching of $V_s$, the voltage on the capacitor initially rises, then snaps back after which the capacitor resumes a typical charging behavior. The data are replotted as charge density σ vs. voltage $V_f$ in Fig. 4d, and the negative slope $\left(\frac{d\sigma}{dV_f}\right)$ is identified as differential negative capacitance density shown in Fig. 4d inset. We tested the H-NNO capacitor with different resistors and all exhibited the NDC effect (Fig. 4e). We also tested our setup by replacing the H-NNO capacitor with a commercial dielectric capacitor of 529 pF and the results (Fig. S12) indicate that the NDC effect in H-NNO is robust. The physical origin of transient negative differential capacitance is the mismatch in the switching rate between the free charge on the metal plate and the bound charge in a ferroelectric (FE) capacitor during the polarization switching[41,42]. The four regions are shown schematically in Fig. 4f, labeled "①" to "④" and are also marked in Fig. 4a, c. When the external positive charges start flowing onto the FE capacitor, the net electric field $\vec{E}$ or electrical

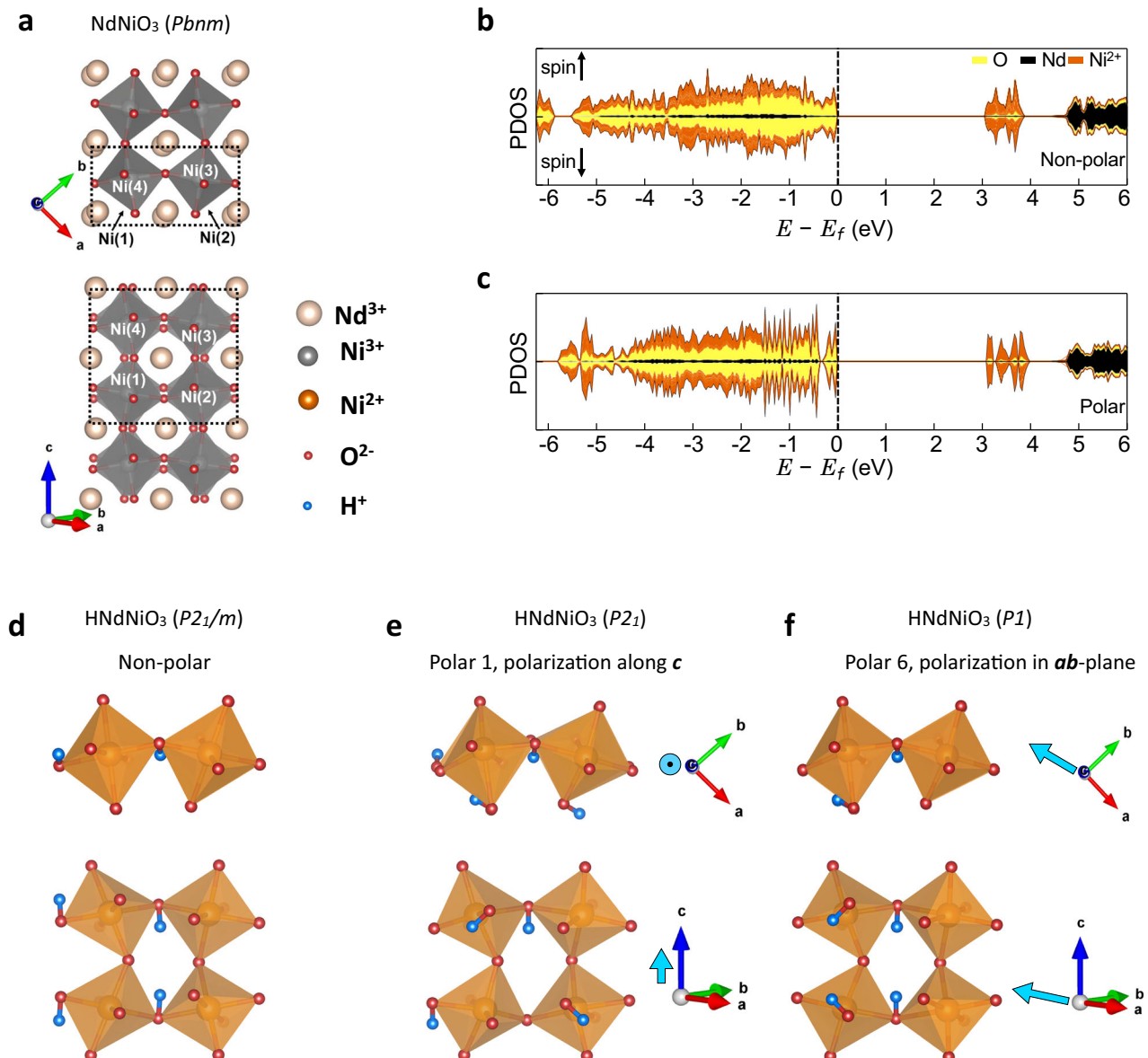

**Fig. 3 | First-principles calculations of hydrogen-doped NdNiO₃. a** The crystal structure of NNO. **b, c** The spin-polarized projected density of states of H-NNO for the non-polar structure as well as the 'Polar 6' structure. **d–f** The crystal structures of H-NNO. **d** Is constructed to maintain the inversion symmetry and is non-polar, while (**e, f**) are polar structures demonstrating polarizations along different crystallographic directions. The positions of the hydrogens can be identified using the bonding to two NiO₆ octahedra. Blue arrows mark the overall direction of the local dipole moment for the polar structures.

potential changes sign. This field consists of two components: $\vec{E}_p$, created by spontaneous polarization, and $\vec{E}_\sigma$, by the surface charge. While $\vec{E}_\sigma$ changes sign, and $\vec{E}_p$ remains in the same direction as the local field is smaller than the FE coercive field. From the moment ② to ③, the voltage drops quickly, i.e., negative differential capacitance region, as the polarization switching rate is faster than the surface charge supply. When most of the dipoles change the sign, the capacitor resumes the normal charging process from ③ to ④. The transient negative differential capacitance phenomenon is evidence of the existence of a switchable spontaneous polarization in H-NNO.

**Use case of polarization relaxations in artificial neural networks**
Multiple polarization states in ferroelectric capacitors (FeCAP) can be induced by applying voltage pulses of different amplitudes or widths. Such ferroelectric capacitors (FeCAP) are promising candidates as neuronal devices that can be interfaced with a crossbar array

architecture[43]. The transient behavior of our devices enables the natural implementation of the leak functionality in neurons. As proof of principle, we measured the potentiation and depression of the polarization in the FeCAP and designed a neural network hardware-software co-design framework to execute a handwritten digit recognition task on the MNIST dataset. The dataset has 60,000 training images and 10,000 testing images of handwritten digits from 0 to 9 each having a pixel dimension of 28 × 28. Following the network structure in ref. 44, our network comprises 784 input neurons and 100 excitatory neurons. Each of the input pixels is converted to a Poisson spike train with a pre-defined intensity. The network is operated over 100 timesteps and the output spikes are accumulated in every timestep. The neurons with the highest number of spikes are designated to corresponding recognized classes. The network is equipped with lateral inhibition (winning neuron inhibits other neurons in the network), homeostasis (threshold adjustment such that it becomes more difficult to fire for an

over-excited neuron), and Spike-Timing-Dependent Plasticity (STDP) mechanisms[44]. Experimental measurements of our FeCAP devices show an exponential relation between change in polarization ($\Delta P$) and applied voltage (Fig. 5a). Necessary hyperparameters for the network are tabulated in Table S3. Based on the experimental measurements, we calibrated our neural network framework where each input to the neuron, analogous to the applied electric field, aggregates exponentially to the corresponding membrane potential which is analogous to the change in polarization. The membrane potential integration equation is as follows,

$$V_{mem}(t+1) = \left(1 - \frac{1}{\tau}\right)V_{mem}(t) + k_1 x e^{k_2 x} \tag{1}$$

Where $\tau$ represents the decay rate of membrane potential (polarization) and $k_1, k_2$ are fitting constants. The relaxation in polarization is critical to implement the leak functionality in spiking neurons which has been shown to enable robustness and better generalization in brain-inspired algorithm design[45]. Figure 5b shows the learned weight map of the 100 excitatory neurons in the network with a training accuracy of 83.59% and inference accuracy of 83% on the MNIST test images. Moreover, for 400 neurons, the training and inference accuracies are 90.64% and 89% respectively for 2 repeated representations of the training images. The results for different numbers of neurons are comparable to ideal software accuracies reported in ref. 44 demonstrating potential relevance to the design of neural network hardware.

## Discussion

In summary, we report the discovery of tunable and switchable polarization in a perovskite nickelate due to proton doping. Proton doping not only induces a metal-to-insulator transition but also results in a metal-to-ferroelectric phase transition. The relaxations in polarization behavior can be used for the design of neural hardware components. The results open avenues for a new knob - namely light–ion doping - in creating ferroelectric polarization properties not seen in parent compounds. Since several oxides also possess magnetic ordering, the study here provides an interesting route to explore the discovery of new multiferroic materials, paving the way for devices

**Table 1 | The energy per formula unit and polarization for the non-polar H-NNO as well as 6 polar structures**

| Structure | Space group | Energy (meV/f.u.) | Polarization (µC/cm²) |
|---|---|---|---|
| Non-polar | $P2_1/m$ (#11) | 0 | (0.0,0.0,0.0) |
| Polar 1 | $P2_1$ (#4) | −159 | (0.0,0.0,1.9) |
| Polar 2 | $P1$ (#1) | −50 | (−20.6, −5.4, 1.4) |
| Polar 3 | $P1$ (#1) | −145 | (−17.4, 14.8, 3.0) |
| Polar 4 | $P1$ (#1) | −68 | (−23.7, −7.0, 0.8) |
| Polar 5 | $P1$ (#1) | −140 | (−28.0, −26.5, −1.2) |
| Polar 6 | $P1$ (#1) | −165 | (−45.7, −12.9, 0.0) |

The italic symbols represent crystallographic space groups.
The non-polar structure is taken as the zero for both the energy and the polarization to define the branch choice. The polarization is reported as a vector in cartesian coordinates where the (110) direction (defined in an orthorhombic cell) - the epitaxial growth direction - is along the unit vector (0.69,0.72,0).

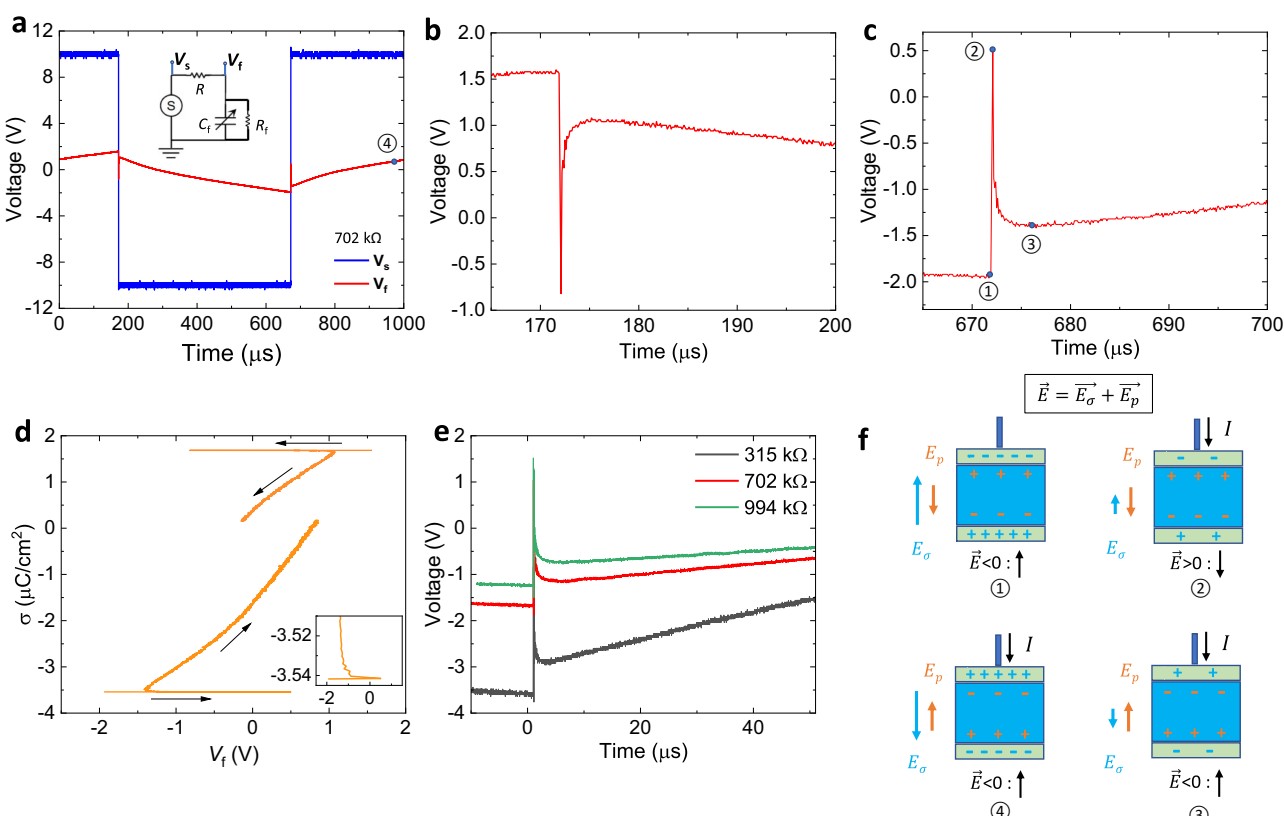

**Fig. 4 | Demonstration of transient negative capacitance. a** Voltage transients of a H-NNO capacitor (1.1 nF at 1 kHz) with a series resistor $R = 702$ kΩ. $V_s$ is the applied voltage and $V_f$ the voltage across the ferroelectric. Inset is the circuit of the pulse measurement. **b, c** The close-up view of NDC regions corresponding to (**a**). **d** Charge density as a function of $V_f$. The negative slope indicates the differential negative capacitance. Inset is the close-up view of the NC region. **e** The negative differential capacitance region with different series resistors. **f** Schematic illustration of switching of spontaneous polarization during a square voltage pulse.

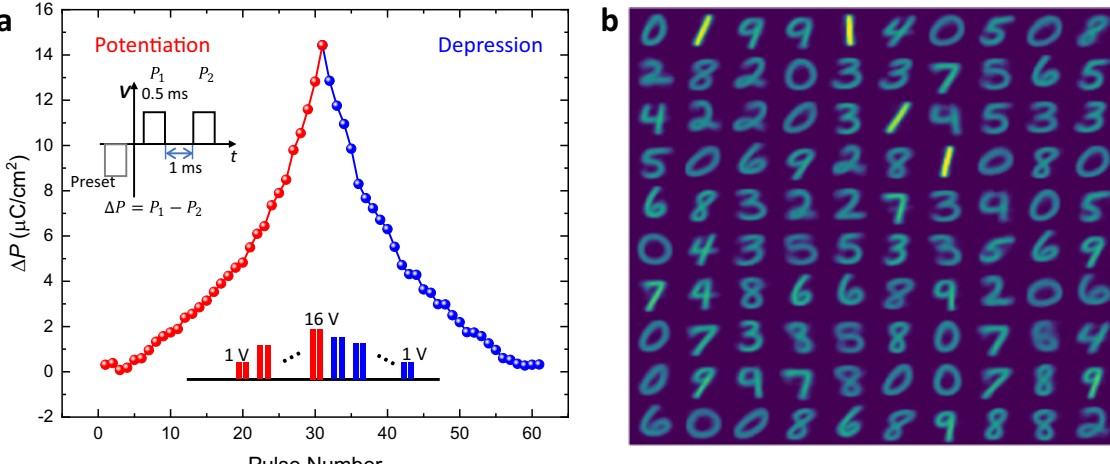

**Fig. 5 | Synaptic characteristics of the H-NNO device and their use in digit recognition. a** The polarization change (ΔP) of the H-NNO device with a series of voltage pulses for potentiation by increasing magnitude, and depression by decreasing magnitude. The pulse sequence is shown in the inset. **b** Trained weight map of the excitatory neuron connections.

with simultaneous electrical and magnetic functions. Further, the proton doping techniques presented here can be readily extended to other material families to examine the resulting polarization behavior.

## Methods

### Synthesis of $NdNiO_3$ and $H\text{-}NdNiO_3$ films

The NNO-based devices studied here consist of a ~150-nm-thick NNO film epitaxially grown with a high-vacuum sputtering system on (001)-oriented conductive $Nb:SrTiO_3$ substrate (Nb doping concentration is 0.5 wt% and the resistivity is $0.003\ \Omega \cdot cm$). The NNO films were deposited by sputtering a NNO ceramic target with radio frequency power at 150 W at room temperature. During the deposition, 10 mTorr pressure of $Ar/O_2$ mixture at 4:1 ratio was used. After the deposition, the film was annealed at 500 °C for 24 h in an ambient atmosphere. Pd electrodes with 50-nm thickness and 150-μm radius were sputtered with a shadow mask onto the NNO film. Pd electrodes serve as a catalyst to incorporate H in the NNO film during annealing in a flowing gas mixture. Moderate doping was performed at 200 °C for 30 minutes under $H_2(5\%)/Ar(95\%)$ mixture gas. Heavy doping was performed at 250 °C for 8 h in pure $H_2$ gas.

### Electrical measurements

The P-E loops and PUND loops were measured with the Ferroelectric Tester (Precision Premier II, Radiant Technologies). For low-temperature measurements from 20 to 300 K, the top Pd electrodes were connected using Ag paint and Ag wires, and a cryostat (Sumitomo Cryogenics) was used. Resistance was determined by measuring current at 0.1 V, 100 ms pulse. To investigate the NDC transients, voltage pulses created by an arbitrary function generator (Tektronix AFG31000 Series) were applied to a series connection of a resistor and the H-NNO capacitor. Voltage transients of the applied pulses and the voltage over the H-NNO capacitor were acquired using an oscilloscope (Tektronix DSOX4154A). All measurements were carried out at room temperature if not explicitly stated.

### Electrical Simulation using COMSOL

The 2D cross-section of the H-NNO device has been simulated with the Electrostatics module of COMSOL Multiphysics. The circular Pd electrode has a radius of 150 μm, thickness of 0.05 μm, and the gap size between the two electrodes is 200 μm. The voltage between the two electrodes is 5 V. The H-NNO film has a thickness of 0.15 μm while the Nb: STO layer is 1 μm. Due to the limitation of the computation capability, the bottom Nb:STO layer has been attached to a floating

potential boundary condition to mimic a 1000 μm thick Nb:STO layer. The overall size of the simulated cross section has a height of 1.2 μm and a length of 1200 μm and is contained in an air box of size 2.2 μm by 1400 μm.

### First-principles calculations of H-NNO

Our first principle DFT + U calculations were carried out using the Perdew-Burke-Enzerhof (PBE) functional[46] as implemented in VASP[47,48] with an energy cutoff of 520 eV and the Nd_3, Ni_pv, O, and H projector augmented wave (PAW)[49] potentials provided with the VASP package. The Nd_3 PAW potential freezes the f-electrons in the core. We include a Hubbard U (within the rotationally invariant method of Liechtenstein et al.[50]) with U = 4.6 eV and J = 0.6 eV following our previous work[51]. All structural relaxations are carried out with Gaussian smearing with = 0.1 eV and a Monkhorst-Pack k-mesh of 6x6x4 such that the forces were less than 0.005 eV/Å. The density-of-states calculations are performed using the tetrahedral method with Blöchl corrections[52]. The projected density-of-states (PDOS) plots are generated using a Γ-centered k-point mesh and the site-projected scheme of pymatgen[53]. The polarization is calculated using the Berry phase method, as implemented in VASP[54]. To ensure that the same branch was used in the polarization calculations, the calculated polarization is tracked over a linearly interpolated path starting from a non-polar structure. We use a 20-atom unit cell of NNO to accommodate the $a^-a^-c^+$ tilt pattern of the *Pbnm* structure. As NNO is either paramagnetic or antiferromagnetic in the temperature range studied here, we do not study ferromagnetic ordering. We choose a G-type antiferromagnetic (AFM) ordering in our spin-polarized DFT calculations as previous work has shown that while the type of AFM ordering affects the bandwidth, it does not affect electron localization with electron doping[51]. After performing a structural optimization of the 20-atom NNO cell, we add 4 H atoms to study H-NNO, we add 4 H atoms to the 20-atom cell and relaxed the cell according to the epitaxial constraint of the experimental set-up using the strained-bulk method only allowing for volume expansion in the direction of epitaxial growth[55].

### X-ray absorption spectroscopy and X-ray photoemission electron microscopy

XAS, XPEEM measurements have been performed at the XPEEM/LEEM endstation of the Electron Spectro-Microscopy beamline (ESM, 21-ID) at the National Synchrotron Light Source II. XAS measurements were performed in a partial yield mode collecting the secondary electrons, with the 2 eV energy analyzer slit centered over the

maximum of the secondary electron emission peak. Pixel-wise XAS was obtained by recording a series of XPEEM images at each energy in each absorption edge range at sequential increments of 0.2 eV. Local XAS spectra can be extracted by plotting the intensity of a given spatial region (set of pixels) at each energy. For example, for Ni *L*-edge, 152 XPEEM images were acquired from a photon energy range of 850 eV–880.2 eV.

### X-ray reflectivity and synchrotron X-ray microdiffraction measurements

Synchrotron XRR and X-ray microdiffraction measurements of the H-NNO samples were carried out on a five-circle diffractometer with χ-circle geometry (in which the sample can be rotated around the center of the diffractometer), using an X-ray energy of 10 keV (wavelength $\lambda = 1.2398$ Å) at beamline 12-ID-D of the Advanced Photon Source of Argonne National Laboratory. The X-ray beam is optically focused by an APS-developed 3D-printed polymer compound refractive lens onto the electrode-patterned sample. The focused X-ray beam profile is 1.3 μm (vertical) by 1.8 μm (horizontal) so it can probe the position-dependent lattice structure due to protonation distribution across the Pd-Pd electrode gap patterned on the $NdNiO_3$ sample. The actual X-ray footprint on the sample is enlarged a few times along the X-ray beam direction due to the angle of incidence. The lateral resolution to differentiate position-dependent lattice structure across the gap is defined by the X-ray beam horizontal profile. A high-resolution small-pixel EigerX 500 K area detector (75 μm pixel size) was used to acquire the microdiffracted X-ray beam from the sample. The diffraction signals were integrated by using a selected region of interest in the 2D area detector images. We first used a larger range 2D mesh scan to map out the electrode gap area of interest. Then we performed a series of XRD scans along a line-cut across the electrode gap to obtain position-dependent lattice structures.

Synchrotron 3D reciprocal space mapping of the NNO and H-NNO samples was carried out on a six-circle Huber diffractometer using an X-ray energy of 17.7 KeV (wavelength $\lambda = 0.7004$ Å) at the beamline 4-ID of National Synchrotron Light Source II of Brookhaven National Laboratory.

### Elastic recoil detection analysis

A 2.3 MeV $He^{2+}$ ion beam of diameter 2 mm in a 1.7 MV IonexTandetron accelerator was used to probe the hydrogen concentration. The beam was oriented with a scattering angle of 163° and an ERDA detector was utilized to record the arrival of H atoms ejected from the sample through elastic collisions. Only forward-scattered H ions were able to penetrate the detector. The probe depth of ERDA is approximately 1 μm.

## Data availability

All relevant data are available within the paper and its Supplementary Information files. Source data are provided with this paper.

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

## Acknowledgements

This research used resources of the Center for Functional Nanomaterials and the National Synchrotron Light Source II, which are U.S. Department of Energy (DOE) Office of Science facilities at Brookhaven National Laboratory, under Contract No. DE-SC0012704. This research used resources of the Advanced Photon Source, a U.S. Department of Energy (DOE) Office of Science user facility operated for the DOE Office of Science by Argonne National Laboratory under Contract No. DE-AC02-06CH11357. The authors acknowledge AFOSR grant FA9550-22-1-0344 for supporting the compositional characterization and polarization studies of the films. K.D. and S.W.C. were supported by the DOE under Grant No. DOE: DE-FG02-07ER46382. We acknowledge the Laboratory for Surface Modification at Rutgers University for ERDA measurements.

## Author contributions

The thin films were synthesized by T.J.P. and H.Y. Y.Y., M.K., T.J.P., and S.R. coordinated the study. Low-temperature measurements were performed by Y.N. under the supervision of X.X. Polarization measurements were performed by Y.Y., T.J.P., S.D., and X.L. X.L. carried out the polarization measurements under the supervision of P.Y. Transient negative differential capacitance was investigated by Y.Y. First-principles calculations were performed by M.K. and K.M.R. Synchrotron XRR and XRD were performed by H.Z. XAS and XPEEM was studied by J.S. and A.A. Lab-based XRD was performed by R.K.P. Piezoresponse force microscopy was performed by K.D. under the supervision of S.C. Rutherford Back-scattering Spectrometry and ERDA data were analyzed by R.B. and R.K.P. Neural network simulation was done by A.S. under the supervision of Ab.S.; COSMOL simulation was done by M.Z. under the supervision of C.W.; Hydrogen concentration effect on polarization was investigated by R.K.P. and Y.Y. Y.Y., T.J.P., M.K., and S.R. co-wrote the manuscript. All the authors discussed the results and commented on the manuscript.

## Competing interests

The authors declare no competing interests.
