## [Peer Review File · Nature Communications]

Hydrogen-induced tunable remanent polarization in a perovskite nickelateREVIEWER COMMENTS

Reviewer #1 (Remarks to the Author):

Yuan et al. reported the tunable remanent polarization in perovskite nickelate by H doping. The hydrogen doping leads a phase transition from metallic phase to insulating states with a polarization. The authors carefully characterized the films and electrical properties, and proposed the possible temporal regime for neural network. These are interesting results. However, the electrical properties are transient, only exist within 1 second. In my opinion, it is useless. Furthermore, though the phase transition sounds interesting, the origin is not clear. The paper includes some first principles calculations, which shows an insulating pristine NNO, however it is metallic in experiments. I do not recommend it to be published in Nature communications.

Reviewer #2 (Remarks to the Author):

The authors report the observation of a remanent polarization in NdNiO₃ (NNO) films after the introduction of hydrogen (H-NNO). At room temperature, NNO is metallic and undergoes a metal-insulator transition at low temperatures. After hydrogen-doping, the films become insulating. The resulting proton-doped insulating films display both dipolar polarization and space-charge polarization. The polarization relaxes with 1s, and it is switchable and tunable. The authors also explore the application of the system in artificial neural networks. The discovery of switchable polarization in doped perovskite nickelates is a noteworthy finding. This is especially significant as ferroelectricity was one of the properties that had not yet been observed in this family of compounds. However, the technical writing style and frequent transitions between several panel figures in the main manuscript and supporting material make the paper less accessible to a broad readership of Nature Communications. Before considering publication in Nature Communications, it is important to make a concerted effort in this direction.

Additional comments at this stage:

The 150 nm thick NNO films grown on SrTiO₃ are likely relaxed or partially relaxed. Why a strained structure is considered in the DFT is unclear.

The calculations suggest a highly distorted structure in H-NNO layers, which may be difficult to accommodate to SrTiO₃ substrates. How does it compare to experimental films?

Additionally, the microstructure of the H-NNO films is unclear. The x-ray diffraction scans suggest that doping significantly reduces the quality of the film. Additional characterization is necessary.

The transport properties of pristine NNO and H-NNO should be included.

Reviewer #3 (Remarks to the Author):

In this manuscript, authors present a detailed study on the characterization, metallic-ferroelectric phase transition and neuromorphic application of hydrogen doped NdNiO₃ thin films, an important member of the perovskite rare-earth nickelate. As in conductive oxides, the free electrons responsible for the metallic behavior will typically eliminate the ferroelectric polarization, it is rare to find an example or approach to transform the metallic state into a polar state. It is the need of the semiconductor technology to pursue available candidates for

more functionalities and future applications. This work is significantly relevant for the still-to-grow field of perovskite ferroelectrics and neuromorphic computing, and the relaxation of polarization for leak functionality in neurons sounds potentially very useful. Its publication should encourage the authors themselves and other researchers to extend such studies to other conductive oxide materials. However, I hope the authors can clarify following concerns before publication.

(1) The NNO films were annealed in H₂/N₂ (5/95) atmosphere. I wonder whether there are abundant oxygen vacancies generated during the annealing under this reduction atmosphere. The oxygen vacancies usually have significant impacts on the conductivity of the films.

(2) Is it possible to quantify the doping concentration of protons in the nickelate films? How does the doping concentration affect the polarization?

(3) The H doping was realized by catalytic spillover which caused inhomogeneous H concentrations in the area between the two Pd electrodes. I am curious how does this inhomogeneous doping affect the polarization? Is there any way to do uniform H doping?

Reviewer response letter

We sincerely thank the Reviewers for taking the time to study our manuscript and provide constructive and thoughtful comments. In our revised manuscript, we have thoroughly addressed the comments of the Reviewers by including new data including structural and compositional characterization, new polarization measurements, revised theoretical description and additional appropriate references, as well as careful editing of the figures and manuscript to improve the readability. Point-by-point responses to the Reviewer comments are provided below. We have also attached an itemized list of our changes, and a version of the revised manuscript with changes highlighted.

REVIEWER COMMENTS

Reviewer #1 (Remarks to the Author):

Yuan et al. reported the tunable remanent polarization in perovskite nickelate by H doping. The hydrogen doping leads a phase transition from metallic phase to insulating states with a polarization. The authors carefully characterized the films and electrical properties, and proposed the possible temporal regime for neural network. These are interesting results. However, the electrical properties are transient, only exist within 1 second. In my opinion, it is useless. Furthermore, though the phase transition sounds interesting, the origin is not clear. The paper includes some first principles calculations, which shows an insulating pristine NNO, however it is metallic in experiments. I do not recommend it to be published in Nature communications.

Response: We thank the reviewer for this evaluation of our study and highlighting '*the authors carefully characterized the films and electrical properties*'. We appreciate the comments and provide further clarification on the importance of transient, i.e. short-term memory for emerging neuromorphic applications. Historically, the use of ferroelectrics in memory technology was focused on realization of binary non-volatile memory that relied on switching of polarization states under an electric field. However, in recent years, due to the surging interest in non-Von Neumann computing, various forms of memory in semiconductor devices have become important. Particularly, for neuromorphic computing, short-term memory has been determined to be of value from the perspective of learning and forgetting. Spontaneous loss of memory state can also be useful for 'reset' function in artificial neural networks. Therefore, we respectfully point out that metastable polarization has significance in the context of emerging neuromorphic computing. We further elaborate on this aspect below with suitable references from literature. The metastability of polarization and its relaxation provides an attractive temporal property that can be leveraged from an algorithm standpoint for neuromorphic computing. The exponentially decaying behavior of polarization as a function of relaxation time has been demonstrated in 'Polarization relaxation dynamics in H-NNO thin films' section. Moreover, in 'Use case of polarization relaxations in artificial neural networks' section, the vital role of polarization relaxation to implement neuron leaky functionality in a network in the context of an unsupervised learning rule has been investigated. From neuroscience point of view, presence of leak in biological neurons has already been reported in sodium ion channels^{1,2} and synaptic transmission in visual cortex^{3,4}. Spiking neural networks (SNNs) may leverage the bio-plausibility

of leaky characteristics in neurons to control the membrane potential (Polarization) decay as a function of time. The membrane potential update equation can be written as follows,

$$V_{mem}(t + 1) = (1 - \frac{1}{\tau})V_{mem}(t) + k_1 x e^{k_2 x}$$

Here, τ denotes the decay rate of membrane potential, V_{mem} , and k_1, k_2 are fitting constants. x refers to the input of a neuronal layer which is analogous to the applied electric field, E . The exponential term in the above equation reflects the exponential relation between remnant polarization, P_r and electric field as shown in Fig. 2f in the paper. Introduction of leaky models in SNNs can make it robust against noise in spike inputs which is advantageous for mitigating the costly re-training procedures in data augmentation tasks and error amplification during quantization⁵. It is worth mentioning here that metastable dynamics in the timescale of ~secs is actually advantageous since this reduces the hardware latency for computing. This is also in agreement with prior works in neuromorphic computing where various relaxation times have been observed in other technologies and used for various applications like 1ms in a memristor device⁶ and 0.5s in an Ag₂S inorganic synapse⁷.

The origin of metal-to-ferroelectric phase transition arises from proton doping and causes the polar structures as also noted from first principles calculations. As to the question, “The paper includes some first principles calculations, which shows an insulating pristine NNO, however it is metallic in experiments”: the DFT calculations are carried out at 0 K, which is why we find an insulating state as there is an insulator-to metal transition around 200K in pristine NNO. We have updated Fig. 3 to avoid any confusion to the reader.

References:

1. Snutch, T. P. & Monteil, A. The Sodium ‘Leak’ Has Finally Been Plugged. *Neuron* vol. 54 Preprint at <https://doi.org/10.1016/j.neuron.2007.05.005> (2007).
2. Ren, D. Sodium leak channels in neuronal excitability and rhythmic behaviors. *Neuron* vol. 72 Preprint at <https://doi.org/10.1016/j.neuron.2011.12.007> (2011).
3. Artun, Ö. B., Shouval, H. Z. & Cooper, L. N. The effect of dynamic synapses on spatiotemporal receptive fields in visual cortex. *Proc Natl Acad Sci U S A* **95**, (1998).
4. Millman, D., Mihalas, S., Kirkwood, A. & Niebur, E. Self-organized criticality occurs in non-conservative neuronal networks during ‘up’ states. *Nat Phys* **6**, (2010).
5. Chowdhury, S. S., Lee, C. & Roy, K. Towards understanding the effect of leak in spiking neural networks. *Neurocomputing* **464**, 83–94 (2021).
6. Chang, T., Jo, S. H. & Lu, W. Short-term memory to long-term memory transition in a nanoscale memristor. *ACS Nano* **5**, (2011).
7. Ohno, T. *et al.* Short-term plasticity and long-term potentiation mimicked in single inorganic synapses. *Nat Mater* **10**, (2011).

Reviewer #2 (Remarks to the Author):

The authors report the observation of a remanent polarization in NdNiO₃ (NNO) films after the introduction of hydrogen (H-NNO). At room temperature, NNO is metallic and undergoes a metal-insulator transition at low temperatures. After hydrogen-doping, the films become insulating. The resulting proton-doped insulating films display both dipolar polarization and space-charge polarization. The polarization relaxes with 1s, and it is switchable and tunable. The authors also explore the application of the system in artificial neural networks.

The discovery of switchable polarization in doped perovskite nickelates is a noteworthy finding. This is especially significant as ferroelectricity was one of the properties that had not yet been observed in this family of compounds. However, the technical writing style and frequent transitions between several panel figures in the main manuscript and supporting material make the paper less accessible to a broad readership of Nature Communications. Before considering publication in Nature Communications, it is important to make a concerted effort in this direction.

Response: We sincerely thank the referee for their positive evaluation of our study and constructive comments. To make the work more readable, we have minimized the frequency of transitions between panel figures in the main manuscript and supporting information and streamlined the flow of information. The figures in the supplementary information have been reorganized accordingly. Clearer explanations and relevant information for figures are also included in the figure captions to improve clarity. We believe that our improved work will be more accessible to a broad readership of Nature Communications.

Additional comments at this stage:

The 150 nm thick NNO films grown on SrTiO₃ are likely relaxed or partially relaxed. Why a strained structure is considered in the DFT is unclear. The calculations suggest a highly distorted structure in H-NNO layers, which may be difficult to accommodate to SrTiO₃ substrates. How does it compare to experimental films?

Response: Thank you very much for the valuable comments. The first principles calculations in fact started from relaxed NNO but allowed to expand in the direction normal to surface. Previous works have shown the H-doping leads to an out-of-plane expansion of NNO films^{8,9}, which is the growth direction. Our experimental XRD and RSM results indicate that H-doping increases the out-of-plane lattice constant (c_{pc}) and the in-plane lattice constant (a_{pc}) remains similar due to the substrate constraint. The calculation shows the increase of the out-of-plane lattice parameter and the volume which agrees with XRD and RSM results.

To improve clarity, we modified the statement in the manuscript,

"The structure was first optimized while accommodating the experimental epitaxial constraint, finding that the structural symmetry in NNO is lowered from Pbnm (space group #62) to P21/m (space group #11) by the epitaxial constraint (Fig. 3a)."

and the following in the methods: "To study H-NNO, we add 4 H atoms to the 20-atom cell and relax the cell according to the epitaxial constraint of the experimental set-up using the strained-bulk method."

to:

"The structure with added hydrogen was optimized while accommodating the experimental epitaxial constraint, i.e. only allowing the bulk lattice parameters of NNO to expand in the direction of epitaxial growth. For a non-polar structure, we find that the structural symmetry in NNO is lowered from Pbnm (space group #62) to P21/m (space group #11) by the epitaxial constraint (Fig. 3a,e)."

And " After performing a structural optimization of the 20-atom NNO cell, we add 4 H atoms to study H-NNO. This cell is then relaxed according to the epitaxial constraint of the experimental set-up using the strained-bulk method only allowing for volume expansion in the direction of epitaxial growth."

All changes in the manuscript are highlighted in yellow.

References:

8. Chen, H. *et al.* Protonation-Induced Colossal Chemical Expansion and Property Tuning in NdNiO₃ Revealed by Proton Concentration Gradient Thin Films. *Nano Lett* **22**, 8983–8990 (2022).
9. Gao, L. *et al.* Unveiling Strong Ion–Electron–Lattice Coupling and Electronic Antidoping in Hydrogenated Perovskite Nickelate. *Advanced Materials* 2300617 (2023).

Additionally, the microstructure of the H-NNO films is unclear. The x-ray diffraction scans suggest that doping significantly reduces the quality of the film. Additional characterization is necessary.

The transport properties of pristine NNO and H-NNO should be included.

Response: Thank you for the suggestions. To illustrate more on the microstructure of H-NNO films, we performed new XRD experiments and synchrotron reciprocal lattice mapping (RSM) on both NNO and H-NNO films. The results are shown below in Figure 1f, g and Figure S2. We have confirmed that the pristine NNO film is oriented well on the Nb:STO substrate. The in-plane strain is tensile for the NNO film with ϵ_{\parallel} of +1.54%. The structural characteristics of our NNO films on Nb:STO are in agreement with previous reports.^{10,11} The XRD spectra in Fig. S2 show the NNO (002)_{pc} peak begins to overlap with the substrate peak by increasing the doping level. It is observed that the (002)_{pc} peak position shifts towards a lower diffraction angle with the injection of protons, implying that the crystal lattice expands along out-of-plane direction (c-axis). The results are consistent with literature reports on H-NNO/STO films^{12,13} and H-NNO/LAO films^{8,9}.

The synchrotron RSM around (112) diffraction peaks indicates that the H-doping in NNO on Nb:STO increases the out-of-plane lattice constant c_{pc} by 1.2% and the in-plane lattice constant a_{pc} remains almost unchanged after hydrogenation in our sample. The preferential change in c_{pc} is attributed to the substrate constraint along in-plane direction and the unrestricted out-of-plane growth direction. The results on the structural change have been noted in the report by Sidik *et al.*¹¹ They indicated that the in-plane lattice constant a_{pc} in epitaxial H-NNO films is clamped by various substrates, such as KTO, STO, LSAT, and LAO.

Details about the structural characterization are described in the revised manuscript and highlighted in yellow.

Figure 1 Synchrotron Reciprocal Space mapping (RSM) around (112) diffraction peaks of (f) pristine NNO and (g) hydrogenated H-NNO films on Nb:STO substrates. The hexagrams mark the (112)_{pc} peak positions of the films, which are determined by fitting the contours. Panel (f) and (g) use the same color bar which is the logarithm of the intensity.

Figure S2 θ - 2θ Laboratory XRD curves with Cu $K\alpha$, β source for NbSTO substrates, pristine NNO/NbSTO, moderately doped H-NNO/NbSTO, and heavily doped H-NNO/NbSTO. Moderate doping was performed at 200 °C for 30 minutes under $H_2(5\%)/Ar(95\%)$ mixture gas. Heavy doping was performed at 250 °C for 8 hours in pure H_2 gas. All film thicknesses are around 100 nm. The dash line marks the change of the (002)_{pc} peak position in the film with the doping levels. The (002)_{pc} peak shifts towards a low diffraction angle and begins to overlap with the substrate peak.

For the transport properties of the NNO and H-NNO films, the current-voltage characteristics for the Pd/NNO/NbSTO device and a series of Pd/H-NNO/NbSTO devices with different doping conditions were studied as shown in Supplementary Figure S13b, indicating that the resistivity increases with the increase in proton doping concentration and the change is up to five orders of magnitudes. The results are consistent with literature.¹⁴

Figure S13 (b) Room temperature current-voltage (I - V) characteristics for the Pd/NNO/NbSTO device and a series of Pd/H-NNO/NbSTO devices with different doping levels, plotted on a semi-log plot. The resistance which is calculated by fitting from 0 to 0.1 V, increases from 752 to $4.6 \times 10^7 \Omega$ by proton doping.

References:

8. Chen, H. *et al.* Protonation-Induced Colossal Chemical Expansion and Property Tuning in NdNiO₃ Revealed by Proton Concentration Gradient Thin Films. *Nano Lett* **22**, 8983–8990 (2022).
9. Gao, L. *et al.* Unveiling Strong Ion–Electron–Lattice Coupling and Electronic Antidoping in Hydrogenated Perovskite Nickelate. *Advanced Materials* 2300617 (2023).
10. Heo, S., Oh, C., Son, J. & Jang, H. M. Influence of tensile-strain-induced oxygen deficiency on metal-insulator transitions in NdNiO_{3-δ} epitaxial thin films. *Sci Rep* **7**, (2017).
11. Sidik, U. *et al.* Tunable Proton Diffusion in NdNiO₃ Thin Films under Regulated Lattice Strains. *ACS Appl Electron Mater* **4**, (2022).
12. Ren, H. *et al.* Controllable Strongly Electron-Correlated Properties of NdNiO₃ Induced by Large-Area Protonation with Metal-Acid Treatment. *ACS Appl Electron Mater* **4**, (2022).
13. Wang, Q. *et al.* Strain-Induced Uphill Hydrogen Distribution in Perovskite Oxide Films. *ACS Appl Mater Interfaces* **16**, (2024).
14. Sidik, U., *et al.* Catalytic Hydrogen Doping of NdNiO₃ Thin Films under Electric Fields. *ACS Appl Mater Interfaces* **12**, 54955–54962 (2020).

Reviewer #3 (Remarks to the Author):

In this manuscript, authors present a detailed study on the characterization, metallic-ferroelectric phase transition and neuromorphic application of hydrogen doped NdNiO₃ thin films, an important member of the perovskite rare-earth nickelate. As in conductive oxides, the free electrons responsible for the metallic behavior will typically eliminate the ferroelectric polarization, it is rare to find an example or approach to transform the metallic state into a polar state. It is the need of the semiconductor technology to pursue available candidates for more functionalities and future applications. This work is significantly relevant for the still-to-grow field of perovskite ferroelectrics and neuromorphic computing, and the relaxation of polarization for leak functionality in neurons sounds potentially very useful. Its publication should encourage the authors themselves and other researchers to extend such studies to other conductive oxide materials. However, I hope the authors can clarify following concerns before publication.

(1)The NNO films were annealed in H₂/N₂ (5/95) atmosphere. I wonder whether there are abundant oxygen vacancies generated during the annealing under this reduction atmosphere. The oxygen vacancies usually have significant impacts on the conductivity of the films.

Response: We sincerely thank the referee for their positive evaluation of our study and thoughtful comments. We have performed additional experiments to address the comments.

To evaluate whether there are abundant oxygen vacancies generated during the annealing in hydrogen, we annealed three Pd/NNO/NbSTO samples at different temperatures. The low and moderate doping were performed at 150 °C and 200 °C respectively for 30 minutes under H₂(5%)/Ar(95%) mixture gas. More extensive annealing (SH) was performed at 250 °C for 8 hours in pure H₂ gas (99.999% Purity).

We performed Rutherford backscattering spectrometry (RBS) on our annealed films. As shown in the below Figure 1, the overlapping oxygen peak profiles for different samples suggest there are no significant compositional differences among the samples within the detection limit of the technique.

Further, it is known from literature, the resistivity of NNO increases with increasing oxygen vacancy concentration.¹⁵ The formation of oxygen vacancy in nickelate thin films depends on the oxygen partial pressure and annealing temperature. Previous work¹⁶ indicates that in a low oxygen partial pressure environment [$p(\text{O}_2) < 10^{-20}$ atm], the temperature threshold necessary for oxygen vacancy formation which induces significant increase in the electrical resistivity of perovskite nickelate films is ~300 °C. In this work, the annealing temperature is ~< 250 °C and oxygen partial pressure is around 10⁻⁶ atm, so it is likely the temperature is not high enough to form significant oxygen vacancies to appreciably change the film conductivity.

Figure 1 RBS spectra taken from Pd/NNO/NbSTO samples before annealing (pristine) and after hydrogen annealing at 150, 200, 250 °C.

References:

15. Chandra, M., Das, S., Aziz, F., Prajapat, M. & Mavani, K. R. Induced metal-insulator transition and temperature independent charge transport in NdNiO₃- δ thin films. *J Alloys Compd* **696**, (2017).
16. Kotiuga, M. *et al.* Carrier localization in perovskite nickelates from oxygen vacancies. *Proc Natl Acad Sci U S A* **116**, (2019).

(2) Is it possible to quantify the doping concentration of protons in the nickelate films? How does the doping concentration affect the polarization?

Response: We thank the referee for this question. To address this point, we have performed elastic recoil detection analysis (ERDA) on our H-NNO samples with different annealing conditions to quantify the H concentration. ERDA is a powerful forward scattering ion beam technique that is highly sensitive to light elements particularly hydrogen that is present in a solid. In Figure S13a, the ERDA spectra show the hydrogen content increases with temperature as expected from activated diffusional process. The averaged H concentrations extracted from the experimental data are given in Table S4.

The PUND method was used to measure the remanent polarization for different amplitudes of the applied voltage for three different doped samples at room temperature. Figure S13c shows the dependence of the extracted remanent polarization on the amplitude of the applied voltage V_{\max} in H-NNO samples of different doping levels. At the same voltage, the induced polarization P_r decreases with the proton doping level. This can be explained by considering the gradually changing proton concentration along the depth direction due to the diffusion process. Higher hydrogenation temperature results in smaller proton

concentration gradient which corresponds to smaller resistance gradient. With the same voltage applied across the thin film, the electric field is smaller and therefore induces less polarization in H-NNO.

The new results are added in the Supplementary Note 3.

Figure S13 (a) ERDA spectra of H-NNO samples doped at 150 °C (SL), 200 °C (SM), and 250 °C (SH) respectively. The energy per channel is 0.67 keV. The thickness of H-NNO is around 100 nm. (b) Room temperature current-voltage (I - V) characteristics for the Pd/NNO/NbSTO device and a series of Pd/H-NNO/NbSTO devices with different doping levels. The resistance which is calculated by fitting from 0 to 0.1 V, increases from 752 to $4.6 \times 10^7 \Omega$ by proton doping. (c) Remanent polarization measured by the PUND method as a function of the amplitude of the applied voltage for H-NNO samples doped at different temperatures. The highest doping temperature results in the largest resistivity. The device setup is shown in the inset. Due to the conductive NbSTO substrate, the polarization was induced by the out-of-plane electric field.

Table S4 Parameters obtained from ERDA analysis of ~100 nm thick NNO and H-NNO films.

Sample	NNO	SL, doped at 150 °C	SM, doped at 200 °C	SH, doped at 250 °C
H/Ni Atomic Ratio	0.07	0.1	0.17	0.33

(3) The H doping was realized by catalytic spillover which caused inhomogeneous H concentrations in the area between the two Pd electrodes. I am curious how does this inhomogeneous doping affect the polarization? Is there any way to do uniform H doping?

Response: As we are using a conducting substrate, most of the polarization is induced in the doped areas proximal to the top electrode as shown in Figure 2a (COMSOL-based simulation of the field distribution). It is therefore reasonable that the inhomogeneous distribution further away from the Pd electrode does not significantly affect the polarization. In future work, we will aim to investigate the effect of dopant distribution via different electrode topologies by exploring other doping techniques such as hydride powder annealing that can dope H uniformly.¹⁷

References:

17. Amarasinghe, D. K. *et al.* Electron doping of NdNiO₃ thin films using dual chamber CaH₂ annealing. *J Solid State Chem* **315**, (2022).

We believe that based on the detailed suggestions of the Reviewers, we have substantially improved both the technical depth and the clarity of the manuscript and increased the overall relevance of this paper to the readership of Nature Communications.

List of changes in revised manuscript:

- P3-4: New results about X-ray diffraction (XRD) measurements and reciprocal space mapping (RSM) are added.
- P6: Clarified description of density-functional-theory (DFT) calculations for H-NNO structures.
- P8: Streamline the conclusions. Clarified description of the method of “Synthesis of NdNiO₃ and H-NdNiO₃ films”.
- P9: Clarified description of the method of “First-principles calculations of H-NNO”.
- P10: Add description for RSM in the “Methods” section
- Figure 1: Added synchrotron reciprocal space mapping (RSM) results.
- Figure 3: Removed the PDOS figure for NNO.
- Supplementary information: all the figures are organized in a new sequence.
- Add Supplementary Figure S2 for lab-based XRD results on a series of H-NNO films with different doping levels.
- Add Supplementary Table S1 for RSM and lattice constant results.
- Add Supplementary Figure S3 for lattice constants of NNO and H-NNO films.
- Supplementary Figure S5 caption: Added more description for the PUND procedures.
- Supplementary Note 1: added description for the device circuit model.
- Supplementary Figure S7 caption: Added more description for the panel (a).
- Supplementary Figure S11 caption: Removed “NNO”.
- Added Supplementary Note 3 - The effect of proton doping concentration on the polarization. Added ERDA data to show the proton concentrations, transport characteristics for the H-NNO samples with different doping levels, and remanent polarization dependence on the amplitude of applied voltage for a series of H-NNO samples.
- Edited a few sentences / phrases for correcting grammar and improving clarity

Sincerely,

Yifan Yuan and co-authors

REVIEWERS' COMMENTS

Reviewer #1 (Remarks to the Author):

I still think the main results of the reported transient polarization are merited and there is no novelty. Under electric field, the weak bonded hydrogens will change their alignment and result in a transient polarization, there is nothing special and everything is clear and simply. There is no reason the these results can be published in Nature communications.

Reviewer #2 (Remarks to the Author):

The authors have addressed all my comments and improved the readability. I therefore recommend the manuscript for publication in Nature Communications.

Reviewer #3 (Remarks to the Author):

The authors have carefully addressed the concerns I raised and revised the manuscript appropriately. I therefore support the publication of this manuscript in current form.

Response to Reviewers' Comments

We appreciate the reviewers' valuable comments and constructive suggestions to our manuscript. The remarks of the reviewers are in black, our responses are in blue.

Reviewer #1 (Remarks to the Author):

I still think the main results of the reported transient polarization are merited and there is no novelty. Under electric field, the weak bonded hydrogens will change their alignment and result in a transient polarization, there is nothing special and everything is clear and simply. There is no reason that these results can be published in Nature communications.

RESPONSE: We thank the referee for reading our manuscript files. We sincerely believe the results presented here are non-trivial and interesting and could motivate new directions for discovering functional properties in materials via doping that may not otherwise be found in the pristine parent compounds.

Reviewer #2 (Remarks to the Author):

The authors have addressed all my comments and improved the readability. I therefore recommend the manuscript for publication in Nature Communications.

RESPONSE: Thank you for the recommendation. We are grateful to the referee for studying our revised manuscript files.

Reviewer #3 (Remarks to the Author):

The authors have carefully addressed the concerns I raised and revised the manuscript appropriately. I therefore support the publication of this manuscript in current form.

RESPONSE: Thank you for the recommendation. We are grateful to the referee for studying our revised manuscript files.